# Dietary Choice Reshapes Metabolism in *Drosophila* by Affecting Consumption of Macronutrients

**DOI:** 10.3390/biom12091201

**Published:** 2022-08-30

**Authors:** Olha Strilbytska, Uliana Semaniuk, Volodymyr Bubalo, Kenneth B. Storey, Oleh Lushchak

**Affiliations:** 1Department of Biochemistry and Biotechnology, Vasyl Stefanyk Precarpathian National University, 57 Shevchenka Street, 76018 Ivano-Frankivsk, Ukraine; 2Laboratory of Experimental Toxicology and Mutagenesis, L.I. Medved’s Research Center of Preventive Toxicology, Food and Chemical Safety, MHU, 03680 Kyiv, Ukraine; 3Department of Biology, Carleton University, 1125 Colonel by Drive, Ottawa, ON K1S 5B6, Canada; 4Research and Development University, 13a Shota Rustaveli Street, 76018 Ivano-Frankivsk, Ukraine

**Keywords:** macronutrients, metabolism, *Drosophila*, food choice

## Abstract

The precise regulation of metabolism and feeding behavior is important for preventing the development of metabolic diseases. Here we examine the effects on *Drosophila* metabolism of dietary choice. These changes are predicted to be dependent on both the quantity and quality of the chosen diet. Using a geometric framework for both no-choice and two-choice conditions, we found that feeding decisions led to higher glucose and trehalose levels but lower triglycerides pools. The feeding regimens had similar strategies for macronutrient balancing, and both maximized hemolymph glucose and glycogen content under low protein intake. In addition, the flies showed significant differences in the way they regulated trehalose and triglyceride levels in response to carbohydrate and protein consumption between choice and no-choice nutrition. Under choice conditions, trehalose and triglyceride levels were maximized at the lowest protein and carbohydrate consumption. Thus, we suggest that these changes in carbohydrate and lipid metabolism are caused by differences in the macronutrients consumed by flies. Food choice elicits rapid metabolic changes to maintain energy homeostasis. These results contribute to our understanding of how metabolism is regulated by the revealed nutrient variation in response to food decisions.

## 1. Introduction

Environmental factors and internal needs determine feeding behavior in response to food availability and quality. All living organisms carefully regulate their macronutrient intake to maintain energy balance and maximize various fitness traits. Being a tractable model, *Drosophila* has been extensively used to evaluate the interventions, factors and molecular mechanisms that regulate food ingestion and metabolism [1,2,3,4,5] via specific regulatory pathways including TOR and insulin [6,7,8,9]. However, little is known about how dietary regimes influence feeding decision to regulate physiology and metabolism.

Fruit flies, *Drosophila melanogaster*, share similarities in the organization and functioning of their sensory systems to that of humans and other mammals, and therefore, they are a good model for studying feeding behavior and its subsequent effects on metabolism. *Drosophila* use taste receptors to search for food sources, mating partners and breeding grounds [10]. Fruit flies, and insects in general, are characterized by a broadly dispersed gustatory system, with taste sensory structures located on legs and wings [11]. The taste sensitivity system allows flies to sense food sources without direct contact with the feeding apparatus. The Gustatory receptor (Gr) protein family is expressed in different taste cells within the *Drosophila* body [11]. Some other receptors were shown to regulate taste in fruit flies including members of a large gene family encoding Ionotropic receptor (Ir), and more specifically, genes of Transient Receptor Potential (TRP) and pickpocket (ppt) [12]. Members of this family have been implicated in fly taste sensing via multidendritic neurons. The neuromodulatory subsets in the decision ensemble were investigated to estimate environmental conditions and internal hunger state. The activity of AstA neurons influences relative carbohydrate and protein preference [13]. Moreover, sugar and amino acid sensing is realized via DH44 and Lk neurons [14,15]. Fan-shaped body neurons FB16 encode food-related decision-making during a sensory conflict between sweet and bittersweet food choices in *Drosophila* [16]. However, the mechanisms governing the balance between nutrient intake and metabolic expenditure in insects remain poorly understood.

Recently, Lyu and colleagues reported that both diet and food choices can influence metabolism and life span in *Drosophila* [17]. However, these effects were discussed without taking into account compensatory feeding or differences in the consumption of specific macronutrients by flies. Hence, we used geometric framework [18] to generate a response of metabolic variables according to consumed protein and carbohydrate [19]. In the present study, we measured the pools of carbohydrates including glucose, trehalose and glycogen as well as lipids when flies were reared on a diet consisting of both sugar and protein in a mixture or when each macronutrient was given separately. 

## 2. Materials and Methods

### 2.1. Maintenance of Insects

Fruit flies *Drosophila melanogaster* of *Canton-S* strain were obtained from Bloomington Stock Center (Indiana University, Bloomington, IN, USA) and cultured on a standard yeast–molasses medium (7.5% molasses, 5% yeast, 6% corn, 1% agar and 0.18 nipagin as a mold inhibitor) at 25 °C under a 12:12 light:dark cycle. Four-day-old females were collected, kept on a standard medium for 24 h to recover after CO_2_ anesthesia and transferred into 1.5 L demographic cages at density of 150 flies per cage and maintained as single-sex populations. Cages were supplemented with two 25-mL vials filled with 5 mL of experimental medium. 

### 2.2. Experimental Media

Experimental flies were kept under two regimens of nutrition for 30 days. Under choice conditions, yeast and sucrose were provided separately, allowing flies to regulate their macronutrient intake to meet their physiological demands. For no-choice conditions, sucrose and yeast were mixed in a single diet. A total of 9 dietary conditions for both types of nutrition were obtained by combining yeast (3, 6 or 12%) and/or sucrose (3, 6 or 12%). All diets also contained 1.2% agar and 0.18% of nipagin as a mold inhibitor. 

### 2.3. Feeding

The feeding rate was determined by adding FD&C Blue No. 1. Five-day-old flies were kept on experimental medium in groups of twenty flies for fifteen days. Subsequently, flies were allowed to feed for 75 min on an experimental medium supplemented with 0.5% dye. After feeding, each group of flies was immediately frozen in liquid nitrogen. Pre-weighed flies were homogenized in 50 mM potassium phosphate (KPi) buffer (pH 7.5) in a ratio 1:10 (weight:volume) using a Potter–Elvejhem glass homogenizer with subsequent centrifugation of 13,500× *g* for 15 min. Supernatants were diluted (1:1 *v*:*v*) with further centrifugation. The absorbance of samples was measured using Spekol-211 at 629 nm. The dye diluted in 50 mM KPi buffer was used to build a calibration curve. We recalculated the volumes of ingested yeast and sucrose solutions into actual intakes of carbohydrates and protein. We used values for yeast composition of 24% carbohydrate and 45% protein as was described previously [19]. To calculate the amounts of carbohydrate and protein eaten, the volumes consumed were multiplied by the concentrations of solutions (30 mg/mL for 3%, 60 mg/mL for 6% and 120 mg/mL for 12% solutions) [2].

### 2.4. Metabolites Pool

*Canton-S* female flies were kept on experimental medium for 30 days and used to measure concentrations of glucose and trehalose in the hemolymph as well as glucose, trehalose, glycogen and TAG in the body. Weighed flies were decapitated and mixed with 10 mM sodium phosphate buffer (SPB, pH 7.4) in a ratio 1:5 (*w*/*v*) with subsequent centrifugation (3000× *g*, 6 min, 4 °C). Bodies were subsequently homogenized in 10 mM SPB (1:10 *w*/*v*) and centrifuged (16,000× *g*, 15 min). Hemolymph and supernatants were incubated for 5 min at 70 °C for inactivation of all proteins followed by centrifugation (16,000× *g*, 15 min, 4 °C) [5]. Measurements were performed using a glucose assay kit (Liquick Cor-Glucose diagnostic kit, Cormay, Lublin, Poland, Cat. #2-203). Trehalose and glycogen were converted into glucose by the incubation of supernatants with 7.4 µU/µL porcine kidney trehalase (Sigma-Aldrich, St. Louis, MO, USA, #8778) or 1.5 mU/µL amyloglucosidase from *Aspergillus n**iger* (Sigma-Aldrich, #10115) during 18 h at 37 °C. The absorbance of samples was measured at 500 nm with a Specoll-211 spectrophotometer (Karl Zeiss, Jena, Germany). For TAG determination, flies were weighed, homogenized in 200 mM PBST (phosphate buffered saline containing 0.05% Triton X100), boiled for 10 min and centrifuged (13,000× *g*, 10 min). The resulting supernatants were used for TAG assay with the Liquick Cor-TG diagnostic kit (Cormay, Poland). The absorbance of samples was measured at 550 nm, as above.

### 2.5. Data Analysis

Statistical comparisons were performed using GraphPad Prism 8 software and additionally re-assessed in R v.3.3.1. Metabolite levels were analyzed using two-way analysis of variance (ANOVA) to compare the effects of two factors: type of feeding (choice vs. no-choice) and diet composition. Multiple comparisons were conducted following the two-stage linear step-up procedure of Benjamini, Krieger and Yekutieli, with Q = 1%. Each row was analyzed individually using *t*-tests, without assuming a consistent SD, and *p* values of less than 0.05 were accepted as a significant difference between groups. Response surfaces for metabolic variables were fitting onto bi-coordinate intakes of protein and carbohydrate and visualized using the “fields” package in R [20].

## 3. Results

### 3.1. Choice Feeding Affects Metabolites Pools in Drosophila

Body glucose (BG) content depended significantly on the feeding option (ANOVA: *F*_1,48_ = 33.18, *p* < 0.0001) given to *Drosophila*: carbohydrate alone, protein alone or both mixed. In flies given a choice of food, BG was higher than in flies in the no-choice conditions (Figure 1A). Indeed, we found a 17–33% higher level of BG in flies given a choice of medium 3S-3Y, 6S-6Y, 6S-6Y, 12S-3Y, 12S-6Y and 12S-12Y as compared with appropriate no-choice regimens (*p* < 0.005). We found that hemolymph glucose (HG) concentration was independent of the type of feeding of the flies (Figure 1B). However, the concentration of trehalose in the body (BT) depended on the type of feeding (*F*_1,50_ = 331.12, *p* < 0.0001; Figure 1C). Consumption of 3S-3Y, 6S-6Y or 12S-3Y media in a mixture led to ~35% higher BT level compared with the appropriate choice media (*p* < 0.02). The concentration of trehalose in hemolymph (HT) was also dependent on the type of feeding (*F*_1,52_ = 11.81, *p* = 0.0012; Figure 1D). Higher HT levels were found in flies that consumed separated 3S and 6Y media compared with mixed 3S-6Y medium (*p* = 0.002). The glycogen pools in fly were dependent on the type of feeding (*F*_1,54_ = 27.45, *p* < 0.0001), the diet composition (*F*_8,54_ = 5.412, *p* < 0.0001) and the interaction between these two factors (*F*_8,54_ = 3.030, *p* = 0.0069) (Figure 1E). Flies kept under the choice condition and fed 3S-3Y, 6S-6Y or 12S-12Y diets had 1.4–1.8-fold higher glycogen content compared with the appropriate diet under no-choice feeding (Figure 1E; *p* < 0.03). The content of triglycerides (TAG) was dependent on the type of nutrition (*F*_1,54_ = 9.742, *p* < 0.0001), diet composition (*F*_8,54_ = 136.2, *p* < 0.0001) and the interaction of both factors (*F*_8,54_ = 9.875, *p* < 0.0001) (Figure 1F). Flies fed 3S-3Y, 3S-6Y, 6S-3Y, 6S-12Y, 12S-6Y or 12S-12Y mixed diets had approximately 2–5-fold higher TAG content compared with those given a choice (Figure 1F; *p* < 0.009).

### 3.2. Food Intake Is Affected by the Choice Feeding

Next, we examined the nutrient intake in response to the choice feeding regimen across a range of diets that varied in protein, carbohydrate and caloric content. We observed significant effects of dietary choice (*F*_1,36_ = 26.11, *p* < 0.0001), diet composition (*F*_8,36_ = 32.63, *p* < 0.0001) and the interaction of these two factors (*F*_8,36_ = 12.35, *p* < 0.0001) on the amount of protein consumed (Figure 2A). Higher amounts of protein were consumed by females reared on the no-choice diets 3S-6Y, 6S-6Y, 6S-12Y, 12S-3Y and 12S-6Y (Figure 2A; *p* < 0.04). However, we observed a 35% higher amount of protein eaten by females given 3S-12Y separated media compared with mixed medium (*p* = 0.03). Sucrose consumption was significantly dependent on the type of feeding (*F*_1,36_ = 135.0, *p* < 0.0001), food composition (*F*_8,36_ = 69.76, *p* < 0.0001) and their interaction (*F*_8,36_ = 15.27, *p* < 0.0001) (Figure 2B). Significantly higher amounts of sucrose were consumed under 3S-6Y, 6S-6Y, 6S-12Y, 12S-3Y and 12S-6Y mixed diets (*p* < 0.006). These results suggested that flies change their feeding behavior to control their protein intake and that there are significant differences in how flies regulate their feeding with respect to the protein and carbohydrate composition of the diet. To test these hypotheses, we calculated the protein-to-carbohydrate (P:C) ratio of consumed macronutrients (Figure 2C). The significance of the P:C ratio was defined by dietary choice (*F*_1,36_ = 11.30, *p* = 0.0018), food composition (*F*_8,36_ = 76.85, *p* < 0.0001) and their interaction (*F*_8,36_ = 5.361, *p* = 0.0002). Females consumed food with a higher P:C ratio when fed on 3S-12Y, 6S-3Y and 12S-12Y choice medium compared with flies under the no-choice feeding conditions (*p* < 0.07). However, our results indicated that the P:C ratio was higher in flies reared on a 12S-3Y mixed medium compared with the choice medium (*p* < 0.02).

### 3.3. Effects of Protein and Carbohydrate Intake on Metabolites

The nutritional geometric framework was used to evaluate the responses of the studied variables as functions of the amounts of protein and carbohydrate consumed under choice and no-choice conditions. Under choice nutrition, BG was maximized at low protein eaten combined with high carbohydrate intake (Figure 3A) whereas under no-choice nutrition, the maximum values of BG were observed at low carbohydrate/low protein intakes (Figure 3B). Flies on the dietary choice regimen displayed higher BG on average than flies without the possibility to choose. For example, the BG concentration was 9.3 mg/gww in flies given a choice at P:C 2.5:8, whereas in flies fed a no-choice diet, BG was about 7.0 mg/gww. Parametric response surfaces for HG that were fitted over the macronutrient intake demonstrated the highest HG under low-carbohydrate/low-protein intake in flies under both choice and no-choice nutrition (Figure 3C,D). 

Flies given dietary choice displayed the highest BT level when consuming low amounts of carbohydrate and protein (Figure 3E). In contrast, under no-choice conditions, the highest BT level was observed for flies that consumed high amounts of carbohydrates and protein (Figure 3F). About 1.4-fold higher BT levels were found for flies fed a choice diet at 3.5:5 P:C. HG concentration was the highest when flies consumed low amounts of carbohydrates and protein under dietary choice conditions (Figure 3G). The highest concentration of HT was observed in flies on no-choice nutrition that ate a high amount of carbohydrates and a low amount of protein (Figure 3H).

Flies that consumed low amounts of protein accumulated the highest levels of glycogen regardless of sucrose concentration (Figure 3I). A similar pattern for glycogen content was found for the low-protein no-choice diets (Figure 3J). In flies given a choice, the glycogen level was about 19.6 mg/gww and 17.7 mg/gww at P:C 3.5:5 and 2.5:8, respectively, whereas in flies fed no-choice, glycogen was 14.5 mg/gww and 11.1 mg/gww.

The response surfaces of TAG levels differed markedly according to the type of nutrition. The maximum TAG storage was found in flies that consumed low amounts of protein and low amounts of carbohydrate in separate media (Figure 3K). Consumption of a high amount of protein and carbohydrates in mixed media led to an enhanced TAG pool (Figure 3L). About 3.5-fold higher TAG levels were observed under choice feeding at P:C 2.5:8 as compared to no-choice.

## 4. Discussion

In this study, we examined the response of carbohydrate and lipid metabolism to consumed macronutrients under dietary choice conditions. We observed significant differences in consumption of carbohydrates and proteins under choice feeding. These differences caused a rearrangement of metabolism, especially as a response to the same amounts of consumed macronutrients in flies given no-choice feeding. The greatest effects were observed for the levels of TAG that might reflect the importance of carbohydrate consumption for synthesizing and maintaining higher levels of TAG [8]. Our results are in good agreement with the recent data by Lyu and coauthors, who showed that choice feeding reduced triglyceride abundance [17]. Coupling the consumption of certain foods to maintain optimal protein-to-carbohydrate (P:C) balance might be costly to the flies. Moreover, long-term decision-making conditions are metabolically stressful for the flies [17]. Thus, under the same P:C values, we observed an almost two-fold reduction of TAG content in flies given a food choice (Figure 3K,L). 

We found similar response surfaces for hemolymph glucose and glycogen levels between choice and no-choice feeding. An overall increase in hemolymph glucose levels as a response to a low carbohydrate and low protein consumption under both choice and no-choice nutrition was observed. Glycogen amounts were elevated under low protein consumption which was also shown earlier [5,21]. However, the shapes of the response surfaces for body glucose and trehalose, hemolymph trehalose and TAG levels differed markedly between choice and no-choice nutrition. This suggests that food decisions contribute significantly to reshaping metabolism. The restriction of carbohydrate and protein intake has been shown to increase trehalose in hemolymph and body, as well as TAG. Carbohydrate and lipid levels are primarily regulated by the adipokinetic hormone (AKH), a glucagon-like hormone [22], and *Drosophila* insulin-like peptides (DILPs) that display antagonistic properties to AKH [9]. The DILP-AKH axis (insulin-glucagon in mammals) is regulated by neuropeptide Allatostatin A (AstA) in response to changing ratios of sugar and protein within the diet [13]. Consequently, dietary nutrients intake may influence AstA secretion that, in turn, contribute greatly to the balance between DILP and AKH as the most important regulators of metabolism and aging.

Food choice and intake affects the peptidergic response to reshape metabolism [17]. *Drosophila* sulfakinins (DSKs) regulate food consumption and are involved in signaling satiety [23]. DSKs are produced in a set of brain neurosecretory cells (IPCs) that also produce DILPs. Available data suggest that the DILPs and DSKs act synergistically to regulate feeding and metabolism [23]. DILPs also affect neurons expressing the NPF receptor that is involved in feeding decisions [24]. Impaired food choice behavior has also been found in flies with impaired NPF signaling [25]. It was shown previously that feeding behavior changes across development. Fruit fly larvae prioritize protein intake over carbohydrate intake during development [26]. However, adult flies voluntarily consumed more sugar than yeast when subjected to a choice diet [17].

Choice of meals causes rapid metabolic reprogramming by increasing signaling via serotonin 2A receptors in the brain [17]. Lyu and colleagues also found that fruit flies given a choice consume more sugar and live a shorter life than those given a fixed diet of equal proportions of macronutrients [17]. Measurements of metabolite levels revealed that certain intermediates of the tricarboxylic acid cycle (TCA), that can be converted into amino acids, were increased on the sugar-rich choice diet in a 5HT2A-dependent manner [17]. The choice diet caused higher α-ketoglutarate levels suggesting that it is directly involved in the effects observed in food decisions [27]. Moreover, the involvement of some metabolic enzymes including glutamate dehydrogenase has been proven to respond to choice nutrition [28].

Taken together, our results show that metabolic responses to nutritional regimens are a function of the interaction between food decisions, dietary composition and the intake of specific macronutrients. Dietary choices influence carbohydrate and fat metabolism in a diet-dependent manner and switch consumption into higher P:C ratios. These results contribute to our understanding of nutrition-specific metabolism and flexible responses to physiological requirements. Further studies concerning the characterization of genetic variability in response to different nutritional conditions could prove highly useful for understanding the mechanisms underlying metabolic disease and feeding disorders.

## Figures and Tables

**Figure 1 biomolecules-12-01201-f001:**
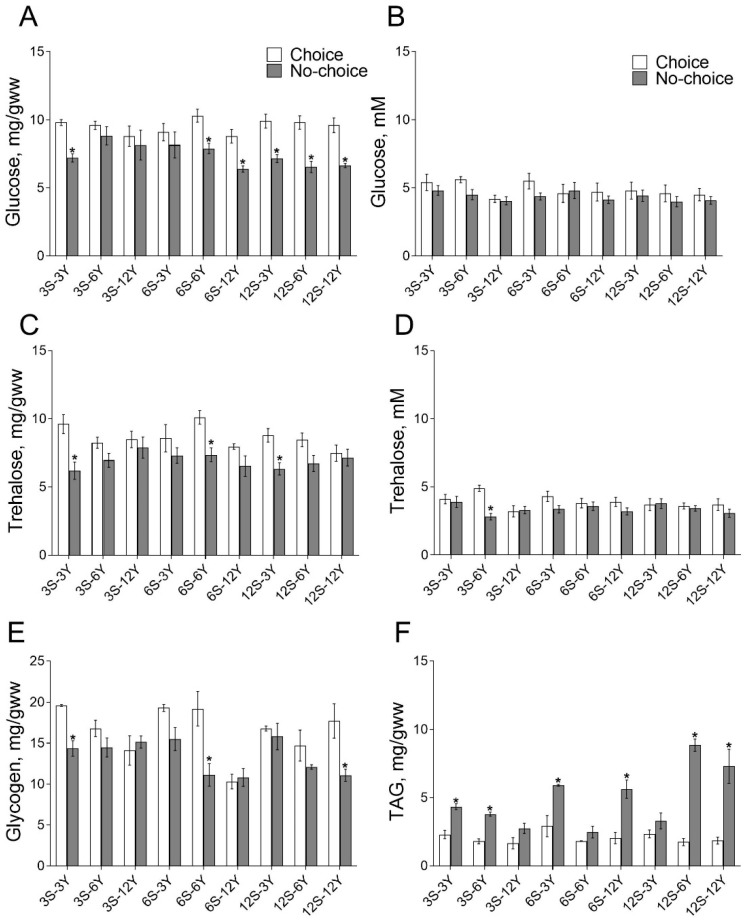
Effects of dietary choice on the levels of metabolites. Body glucose (**A**), hemolymph glucose (**B**), body trehalose (**C**), hemolymph trehalose (**D**), glycogen (**E**) and triglycerides (**F**) under choice and no-choice nutrition. S—sucrose percent in the medium; Y—yeast percent in the medium. Data are presented as mean ± S.E.M. for 4–5 independent replicates, values were considered significantly different if *p* < 0.05 and marked with *. The icons in the subfigures (**A**) also apply to (**C**–**F**).

**Figure 2 biomolecules-12-01201-f002:**
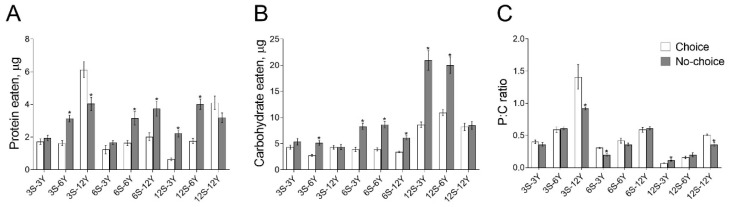
Effects of dietary choice on food consumption. The amount of carbohydrate (**A**) and protein (**B**) intake, as well as protein-to-carbohydrate (P:C) ratio of consumed nutrients (**C**) under choice and no-choice conditions. Data presented as mean ± S.E.M., values considered significantly different if *p* < 0.05 and marked with *. The legend in the subfigure (**C**) also apply to (**A**,**B**).

**Figure 3 biomolecules-12-01201-f003:**
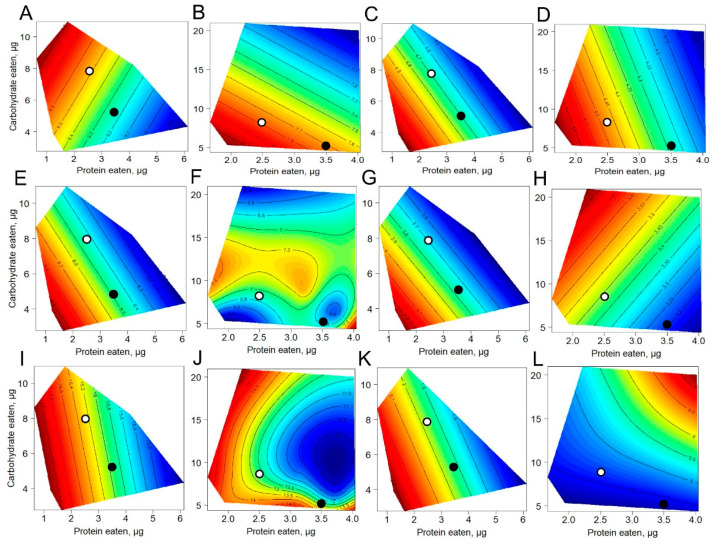
Response surfaces showing dependency of metabolite levels on macronutrients eaten. (**A**) body glucose on choice nutrition; (**B**) body glucose on no-choice nutrition; (**C**) hemolymph glucose on choice nutrition; (**D**) hemolymph glucose on no-choice nutrition; (**E**) body trehalose on choice nutrition; (**F**) body trehalose on no-choice nutrition; (**G**) hemolymph trehalose on choice nutrition; (**H**) hemolymph trehalose on no-choice nutrition; (**I**) glycogen on choice nutrition; (**J**) glycogen on no-choice nutrition; (**K**) triglycerides on choice nutrition; (**L**) triglycerides on no-choice nutrition. Open dots represent the values at P:C of 2.5:8 and filled dots for 3.5:5 P:C ratio. Red color on the response surfaces represents the highest values while dark blue shows the lowest ones, other colors represent intermediate values.

## Data Availability

Not applicable.

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
