# Peer review of "Dietary Choice Reshapes Metabolism in Drosophila by Affecting Consumption of Macronutrients"

_biomolecules, 2022, doi:10.3390/biom12091201_

Round 1

Reviewer 1 Report

Another excellent paper from Lushchak group. Paper is in really good shape and no need to revise

Supplementary opinions:

First of all, I would like to let you know that this is the first manuscript I have given a decision to accept in the current form.

What is the main question addressed by the research? Is it relevant and interesting?
-The authors examined the effects on Drosophila metabolism when flies are subjected to two-choice conditions

How original is the topic? What does it add to the subject area compared with other published material?

-A Recent report from Pletcher group reported that both diet and food
choices can influence metabolism and lifespan in Drosophila [Lyu et al. 2021, eLife]. However, these effects were discussed without taking into account compensatory feeding or differences in consumption of specific macronutrients by flies and this manuscript is original in that it is the first manuscript taking into account those factors.

Is the paper well written? Is the text clear and easy to read?
-Yes, the paper is well written and easy to understand

Are the conclusions consistent with the evidence and arguments presented? Do they address the main question posed?

-The authors  used Geometric Framework to generate a response of metabolic variables according to consumed protein and carbohydrate and the conclusions are consistent with the evidence and arguments presented.

Author Response

Response to Reviewer 1 Comments

Another excellent paper from Lushchak group. Paper is in really good shape and no need to revise

Response: We are very thankful to the reviewer for such a recognition of our work.

Reviewer 2 Report

The authors present a well designed study examining the effect of several defined diets that vary carbohydrate and protein composition under choice and no choice conditions on body and hemolymph metabolite concentrations. Overall, the paper is well written and concise. I have a few minor suggestions that may improve clarity in some places:

Line 70--flies were separated by sex. Please clarify which sex was used in the experiments. Were population cages maintained as single sex populations?

Line 116--specifying the model used for the ANOVA analyses would increase clarity.

Line 118-119--information is given about the t tests, but as written it isn't clear if these analyses are related to the multiple test correction or a separate set of analyses.

Line 120--a brief explanation of the nutritional geometry modeling, response fitting method would increase clarity and interpretability for the naive reader.

Line 139--more attention to the description of the interaction between type of feeding and diet composition on glycogen pools would assist with interpretation, especially because these figures don't look much different from the others described in this section where no interaction is reported.

Figure 1--For this figure and all others, font is awfully small. I appreciate the constraints, but anything to increase readability would be good. I also suggest adding a brief key to abbreviations used in the figure (S and Y).

Figure 3--A brief description of surface colors and interpretation would improve clarity.

Line 220--This sentence is a bit challenging to interpret, especially the second half.

Line 239-252-- and somewhat in following sections, this section feels a bit like a long list of facts. This may benefit from condensing or by drawing clearer insight for the findings reported in the paper.

Author Response

Response to Reviewer 2

The authors present a well designed study examining the effect of several defined diets that vary carbohydrate and protein composition under choice and no choice conditions on body and hemolymph metabolite concentrations. Overall, the paper is well written and concise. I have a few minor suggestions that may improve clarity in some places:

Response: We are very thankful to the reviewer for such a recognition of our work. We have taken the suggestions that helped to significantly improve the manuscript. Introduced changes are marked by blue font in the new version of the manuscript.

Point 1: Line 70--flies were separated by sex. Please clarify which sex was used in the experiments. Were population cages maintained as single sex populations?

Response 1: In new version of the manuscript we have clarified the methods: “Four-day-old females were collected, kept on a standard medium for 24 h to recover after CO2 anesthesia and transferred into 1.5 L demographic cages at density of 150 flies per cage and maintained as single sex populations”

 Point 2: Line 116--specifying the model used for the ANOVA analyses would increase clarity.

Line 118-119--information is given about the t tests, but as written it isn't clear if these analyses are related to the multiple test correction or a separate set of analyses.

Response 2: We appreciate these suggestions and added a more explanation of the statistical procedures.

Point 3: Line 120--a brief explanation of the nutritional geometry modeling, response fitting method would increase clarity and interpretability for the naive reader.

Response 3: Response surface for metabolic variables were fitting onto bi-coordinate intakes of protein and carbohydrate and visualized using “fields” package in R. – We have clarified this in new version of the manuscript.

Point 4: Line 139--more attention to the description of the interaction between type of feeding and diet composition on glycogen pools would assist with interpretation, especially because these figures don't look much different from the others described in this section where no interaction is reported.

Response 4: The presence of the interaction is clearly seen because in most cases there is a decrease of glycogen content when amount of yeast was increased from 3 to 12%. Also in most dietary conditions the values are lower for flies kept under choice conditions. However, we don’t think it has to be described more in section of results.

Point 5: Figure 1--For this figure and all others, font is awfully small. I appreciate the constraints, but anything to increase readability would be good. I also suggest adding a brief key to abbreviations used in the figure (S and Y).

Response 5: We have corrected all these points.

 Point 6: Figure 3--A brief description of surface colors and interpretation would improve clarity.

Response 6: The description has been added.

 Point 7: Line 220--This sentence is a bit challenging to interpret, especially the second half.

Response 7: We appreciate this remark and have rephrased the sentence.

Point 8: Line 239-252-- and somewhat in following sections, this section feels a bit like a long list of facts. This may benefit from condensing or by drawing clearer insight for the findings reported in the paper.

Response 8: We thank a reviewer for these suggestions and have partially rewritten this part of the work.